# Development of a core information set for colorectal cancer surgery: a consensus study

Angus G K McNair [ID] ,[1,2] Robert N Whistance,[1] Barry Main [ID] ,[1] Rachael Forsythe,[3] Rhiannon Macefield,[1] Jonathan Rees,[4,5] Anne Pullyblank [ID] ,[2,6] Kerry Avery [ID] ,[1] Sara Brookes,[7,8] Michael G Thomas,[4] Paul A Sylvester,[4] Ann Russell,[9] Alfred Oliver,[9] Dion Morton,[10] Robin Kennedy,[11] David Jayne,[12] Richard Huxtable,[13] Roland Hackett,[14] Susan Dutton,[15] Mark G Coleman,[16] Mia Card,[4] Julia Brown,[17] Jane Blazeby[1]

For numbered affiliations see end of article.

**Correspondence to**
Mr Angus G K McNair;
angus.mcnair@bristol.ac.uk

## ABSTRACT

**Objective** 'Core information sets' (CISs) represent baseline information, agreed by patients and professionals, to stimulate individualised patient-centred discussions. This study developed a CIS for use before colorectal cancer (CRC) surgery.

**Design** Three phase consensus study: (1) Systematic literature reviews and patient interviews to identify potential information of importance to patients, (2) UK national Delphi survey of patients and professionals to rate the importance of the information, (3) international consensus meeting to agree on the final CIS.

**Setting** UK CRC centres.

**Participants** Purposive sampling was conducted to ensure CRC centre representation based upon geographical region and caseload volume. Responses were received from 63/81 (78%) centres (90 professionals). Adult patients who had undergone CRC surgery were eligible, and purposive sampling was conducted to ensure representation based on age, sex and cancer location (rectum, left and right colon). Responses were received from 97/267 (35%) patients with a wide age range (29–87), equal sex ratio and cancer location. Attendees of the international Tripartite Colorectal Conference were eligible for the consensus meeting.

**Outcomes** Phase 1: Information of potential importance to patients was extracted verbatim and operationalised into a Delphi questionnaire. Phase 2: Patients and professionals rated the importance information on a 9-point Likert scale, and resurveyed following group feedback. Information rated of low importance were discarded using predefined criteria. Phase 3: A modified nominal group technique was used to gain final consensus in separate consensus meetings with patients and professionals.

**Results** Data sources identified 1216 pieces of information that informed a 98-item questionnaire. Analysis led to 50 and 23 information domains being retained after the first and second surveys, respectively. The final CIS included 11 concepts including specific surgical complications, short and long-term survival, disease recurrence, stoma and quality of life issues.

**Conclusions** This study has established a CIS for professionals to discuss with patients before CRC surgery.

## Strengths and limitations of this study

► Robust consensus methodology and transparent reporting was used to allow public scrutiny of all information included or excluded from the core information set.

► Patient and professional views were given equal weighting in the consensus process.

► The patient response rate was low, however, purposive sampling ensured representation of key demographics.

## INTRODUCTION

High-quality, patient-centred communication is a cornerstone of clinical practice in the UK and worldwide.[1–4] Such communication can facilitate shared decision-making, where doctors and patients agree on a treatment plan that best fits the patient's needs, and ensure that patients' fundamental ethical right to self-determination is maintained through the process of informed consent (IC).[5] This is particularly important in surgical oncology. For many patients with cancer, surgery offers the best chance of long-term survival but this must be balanced against short term mortality and potentially irreversible deterioration in quality of life. Understanding and weighing-up the nature and consequences of treatment, based on personal values, is therefore an important part of the cancer journey.

There are, however, significant challenges in addressing the information needs of individual patients. The amount of information that could be discussed is vast, and it is unclear what information is critical to inform understanding in an individual. Generally, patients prefer more rather than less information,[6] but there is a danger of overwhelming patients with information that is not important to

them, and this may reduce understanding and increase anxiety.[7] 'Important' information is itself subjective, and what is important to one person may not be to another. This includes information that may or may not be relevant to the nature and consequences of the proposed surgery. Patient-led communication, where discussions are guided by the individual, is helpful but patients may lack sufficient baseline knowledge to ask important questions.

Beyond these professional and ethical responsibilities, there are additional requirements to meet legal standards of information provision for IC. There has been a gradual shift away from a physician-centred, paternalistic model of consent towards a 'reasonable-patient' standard.[5] This aligns the process of IC to the patients' perspective, and requires surgeons to discuss all relevant information about a proposed treatment that an objective patient would find material to making an informed decision. This model has been endorsed through the recent *Montgomery* ruling in the UK, and is common in other jurisdictions,[8] but it is at once confusing and helpful in addressing patients' information needs.[9] On the one hand, it is a somewhat abstract concept that does little to help physicians tailor information to the individual.[10] Without a clear understanding of who a reasonable patient might be, the temptation exists to over disclose. On the other hand, the reasonable-patient standard might serve best if viewed as a baseline from which more meaningful, person-centred conversations develop.

One method for balancing over and under disclosure of information is to develop a 'core information set' (CIS) for a specific treatment. Core information represents baseline information, determined by patients and clinicians, necessary to stimulate further patient-centred communication.[11–14] A CIS is intended for use once a treatment recommendation or decision has been made and will provide relevant information about a single intervention. This differs from the role and purpose of a decision aid which deliberately provides information about alternative treatment choices.[15] It has the advantages of being feasible and transferable to a wide number of settings, possible to define using established health services research methodology, aligned with contemporary ethical theory, and potentially meeting the 'reasonable patient' legal standard of IC. CISs are available for oesophageal and oropharyngeal cancers.[12 16]

The aim of this study is to define a CIS for colorectal cancer (CRC) surgery. CRC is the third most common cancer in men and second most common in women worldwide, with an estimated incidence over 740 000 and 610 000, respectively.[17] The majority of these patients will undergo a surgical resection and methods are therefore needed to ensure that IC is optimised.

## METHODS
### Study design
This study was conducted in three phases using methods modified from the development of core outcome sets for randomised controlled trials[18]: (1) a long-list of potential information of importance before CRC surgery was identified and categorised into domains, (2) domains were operationalised into a questionnaire that was used to survey stakeholders' views on the importance of each domain using Delphi methods, (3) consensus meetings with patients and professionals were used to finalise the CIS.

This study was conducted in parallel with the development of a set of outcomes to measure in CRC randomised trials, the results of which are published elsewhere.[19] The same study population was used to concurrently address this separate research question. It was hypothesised that there may be observed differences between the core outcome and information sets as they are conceptually different.

### Phase 1: Domain generation
Information about CRC surgery was identified from systematic reviews of clinical and patient reported outcome literature (published elsewhere),[20 21] supplemented by a review of patient information leaflets and interviews with patients to identify additional information not present in the published literature.

An information long-list was created, and similar information was categorised into domains by two members of the study team and a patient representative. The final domains were operationalised into questionnaire items using lay language, and piloted by patients for face validity, understanding and acceptability.

### Phase 2: Delphi consensus process
The questionnaire was sent to key stakeholders (CRC surgeons, specialist nurses and patients who had undergone surgery for CRC). Patients were essential stakeholders as they are the recipient of treatment, and surgeons and nurses have an in-depth understanding of the potential impact of surgery. Family, carers and friends were not included because, although these people form part of clinical discussions, IC is the prerogative of an autonomous individual. Participants were welcome to consult with others when considering the importance of information domains.

Professionals were identified from UK National Health Service hospital trusts that participated in the UK National Bowel Cancer Audit. Non-probabilistic purposive sampling was conducted to ensure centre variation based on geographical region (Northern England, the Midlands, South West and South East England, and Wales), and caseload volume. Patients were recruited from University Hospitals Bristol NHS Foundation Trust, North Bristol NHS Trust and Plymouth Hospitals NHS trust. Participants were approached by post from participating centres and sent a participant information leaflet, a consent form and the questionnaire with a stamp addressed return envelope. Non-probabilistic purposive sampling was conducted to ensure representation based on age, sex and cancer site (rectum, left colon, right

colon). Demographic data, including area of deprivation, marital status, employment status, educational level, were collected. Deprivation was as defined by the UK Office of National Statistics Index of Multiple Deprivation at lower layer Super Output Area level for the individual.[22] This is a combined measure of income, employment, health and disability, education, barriers to public services, crime and living environment. Educational level was defined as up to basic education (to the age of 16 or completion of the UK General Certificate of Secondary Education or equivalent), further education (subsequent qualifications to the age of 18), undergraduate and postgraduate education.

Questionnaires asked participants to rate the importance of information domains on a 9-point Likert scale, ranging from 1 ('not essential') to 9 ('absolutely essential'). Information domains considered not essential after round 1 were discarded (see Data analyses). In round 2, participants were provided with feedback from round 1 in the form of their previous score for each domain and a mean score from their stakeholder group. Participants then rescored each information domain on the 9-point Likert scale, and the results used to determine which domains should be retained and presented in the consensus meetings.

Professionals were also asked to rate the importance of most domains to be included as trial outcome measures in a core outcome set, the results of which are published elsewhere.[19] An example page from the questionnaire is presented in figure 1 to demonstrate how professionals were asked both questions simultaneously. Some information did not have a corresponding trial outcome measure (eg, 'expected in hospital experience' or 'family risk of bowel cancer') and, in those instances, professionals were only asked to rate them as information. Patients were not asked separate questions about trial outcome measures because, based on feedback from the patient steering group, the two concepts were considered synonymous.

### Phase 3: Face-to-face consensus meetings

Separate consensus meetings were held with professionals and patients. The professional consensus meeting was conducted at the Tripartite Colorectal Meeting (meeting of the Association of Coloproctology of Great Britain and Ireland, the American Society of Colon and Rectal Surgeons, the Royal Society of Medicine, Royal Australasian College of Surgeons, Colorectal Surgical Society of Australia and New Zealand and the European Society of Coloproctology) in Birmingham, UK, in 2014. The meeting was open to all members of international societies. The patient meeting was held in Bristol, UK, in 2013. Attendees at this meeting were all from the UK and had completed the questionnaire surveys and responded to an invitation to attend a consensus meeting.

The retained information domains from the second survey were presented and discussed at the meetings. Anonymised voting took place to ask participants to vote each domain as either 'In', 'Out' or 'Unsure' using

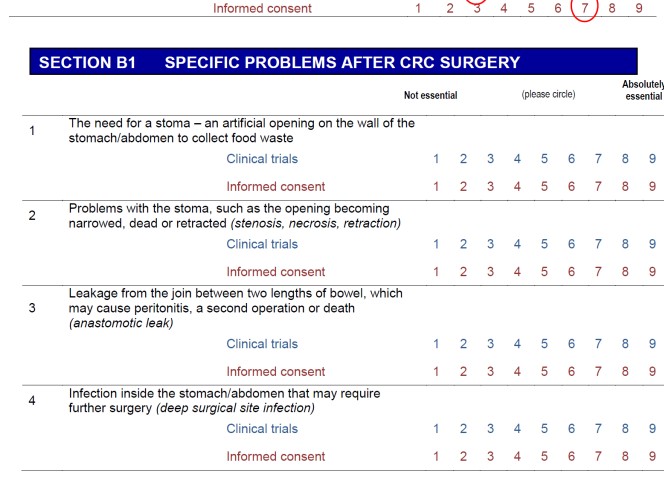

**Figure 1** Questionnaire front sheet. CRC, colorectal cancer.

electronic keypads. Histograms and descriptive statistics were created for each domain during the meeting and displayed to the participants. Where consensus was not reached further discussion ensured and additional voting.

### Sample size

There are no agreed methods to set the sample size for Delphi surveys or consensus meetings. Therefore, an opportunistic approach was used with the aim of obtaining approximately 100 respondents for each stakeholder group for the survey and a smaller group in which discussion could take place in the consensus meetings.

### Data analyses

Information domains rated between 7 and 9 by over 50%, and between 1 and 3 by less than 15%, of respondents in round 1 were retained for round 2. Information domains not meeting these criteria were discarded. Mean scores were calculated for each retained domain to form the feedback for Round 2. Round 2 domains were retained using stricter criteria (between 7 and 9 by over 70%, and between 1 and 3 by less than 15%, of respondents). There are no agreed methods for selecting cut-off criteria within Delphi studies,[23] and therefore the criteria were selected after discussion with collaborators. Domains retained after round 2 were considered in the consensus meetings. Inclusion in the final CIS was determined at the professional consensus meeting by majority vote.

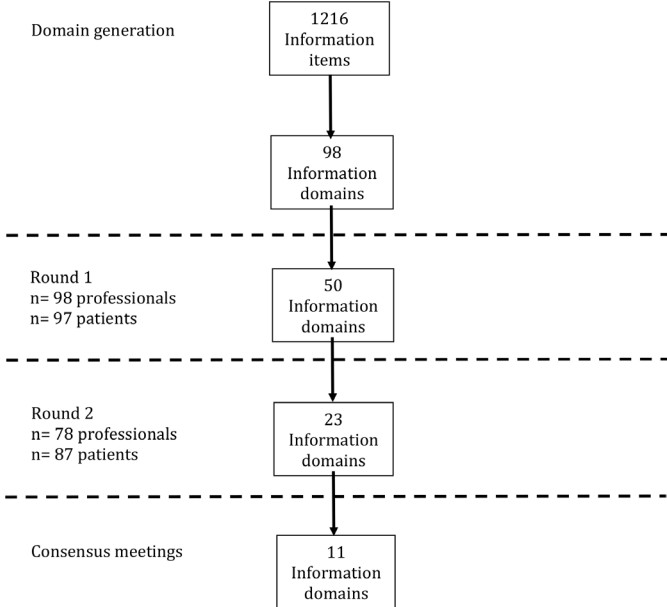

**Figure 2** Summary of results.

In the patient consensus meeting, two rounds of voting were conducted to ensure patients adequately understood the domains. Domains that were voted for by 60% of patients were included in the CIS after initial voting. Those voted for by 40%–60% were brought forward for further discussion and those voted less than 40% were discarded. In the second round, domains that achieved a two-thirds majority were included. All information domains retained from either meetings were included in the final CIS. Patients considered important information for IC and trial outcomes to be synonymous. Results of the patient consensus meeting are therefore presented as part of the core outcome set and are not repeated here.[19]

### Patient and public involvement

Patients collaborators were involved in developing the study design, categorising information from data sources into domains, designing the Delphi questionnaire using patient centred language, and interpreting the clinical significance of results.

### RESULTS
### Phase 1: Domain generation

All data sources identified 1216 pieces of information on CRC surgery that were categorised into 98 domains. A summary of the results is presented in figure 2.

### Phase 2: Delphi consensus process

A total of 63/81 (78%) CRC centres responded including 90 surgeons and 8 specialist nurses (table 1). Centres represented all geographical regions of England and Wales, and caseload ranged from 38 to 275 operations per annum. Patient response rate was 97/267 (36%). The age range was wide, sex ratio equal and similar numbers of patients had rectal, left and right colonic tumours. Many patients lived in areas of low deprivation but there

**Table 1** Participant characteristics

| | Clinical centres | |
| --- | --- | --- |
| | Responders (n=63) | Non-responders (n=18) |
| Region (%) | | |
| Northern England | 14 (22) | 5 (28) |
| Midland | 8 (13) | 0 |
| South East England | 22 (35) | 10 (55) |
| South West England | 9 (14) | 0 |
| Wales | 10 (16) | 3 (17) |
| Mean number of major colorectal resections (range) | 117 (38–275) | 90 (29–210) |

| | Patients | |
| --- | --- | --- |
| | Responders (n=97) | Non-responders (n=170) |
| Mean age (range) | 64 (29–87) | 68 (29–88) |
| Female (%) | 41 (42) | 95 (56) |
| Cancer site (%) | | |
| Rectum/anus | 33 (35) | 55 (32) |
| Left colon | 34 (36) | 46 (27) |
| Right colon | 30 (29) | 60 (36) |
| Unknown | | 9 (5) |
| IMD quintile (%)* | | |
| 1 | 5 (5) | 27 (16) |
| 2 | 13 (13) | 38 (23) |
| 3 | 20 (21) | 24 (14) |
| 4 | 20 (21) | 41 (24) |
| 5 | 39 (40) | 23 (23) |
| Educational level (%) | | |
| Basic | 30 (32) | |
| Higher | 34 (35) | |
| Undergraduate | 16 (16) | |
| Postgraduate | 6 (6) | |
| Not disclosed | 11 (11) | |
| Marital status (%) | | |
| Single/divorced | 17 (18) | |
| Married/cohabiting | 73 (75) | |
| Widowed | 7 (7) | |
| Employment status (%) | | |
| Employed | 16 (17) | |
| Retired | 58 (60) | |
| Seeking work | 1 (1) | |
| Not working voluntarily | 5 (5) | |
| Sickness leave | 5 (5) | |
| Other | 12 (12) | |

Continued

**Table 1** Continued

| | Patients | |
|---|---|---|
| | Responders (n=97) | Non-responders (n=170) |
| Length of hospital stay (%) | | |
| <2 weeks | 80 (83) | |
| 2–3 weeks | 10 (10) | |
| 3–4 weeks | 3 (3) | |
| >4 weeks | 4 (4) | |

*Index of Multiple Deprivation (IMD) as defined by the UK Office of National Statistics at lower layer Super Output Area level for the individual. Lower quintile equates to higher deprivation.

was an even distribution of basic and higher educational level. Professionals rated information about short-term technical outcomes of greatest importance in round 1 including anastomotic leak, adequacy of resection margins and perioperative mortality (online supplementary table 1). Although these issues were also important to patients, a priority was given to information about longer-term outcomes such as survival, distant recurrence and impact on quality of life. A total of 50 domains were retained for round 2.

The response rate in round 2 was 75% (78/104) for health professionals and 90% (87/97) for patients. The provision of feedback and more stringent cut-off criteria in round 2 resulted in 23 domains being retained for consideration in the consensus meetings.

### Phase 3: Consensus meetings

The professional and patient consensus meetings were attended by 61 and 14 participants, respectively. Anonymised voting reached a consensus on six domains at the professionals meeting (table 2). In initial anonymised voting at the patient consensus meeting, 11 domains were voted 'in', three were voted 'out' and nine were inconclusive. Subsequently, domains that were recognised as overlapping were combined.[19] 'Length of hospital stay' was broadened in scope to include all details of patients' pathway through hospital such as the location of the hospital and ward, anticipated length of hospital stay and follow-up arrangements. This information domain was renamed 'Expected in-hospital experiences'. A second round of voting reached a consensus on including three more domains into the CIS: 'anastomotic leak', 'stoma complications' and 'sexual functioning'. Patient and professional CIS were then combined (box 1), creating a final list of 11 domains.

### DISCUSSION

This study developed a CIS to inform consent for CRC surgery. A wide range of sources including published studies and patient interviews were used to identify the

**Table 2** Voting on information domains to be included in the CIS in the surgeon consensus meetings

| | Voted in (n=61) | |
|---|---|---|
| Information domain | N (%) | Consensus |
| Anastomotic leak | 40 (65) | In |
| Conversion to open operation | 38 (63) | In |
| Post-operative mortality | 41 (62) | In |
| DVT/PE | 36 (59) | In |
| Long-term survival | 33 (54) | In |
| Stoma formation | 32 (53) | In |
| Surgical site infection | 30 (49) | Out |
| Cancer recurrence | 29 (48) | Out |
| Reoperation | 26 (43) | Out |
| Haemorrhage | 25 (41) | Out |
| Sexual functioning | 25 (41) | Out |
| Global quality of life | 24 (40) | Out |
| Bowel obstruction | 22 (37) | Out |
| Faecal incontinence | 20 (34) | Out |
| Expected in-hospital experience | 20 (34) | Out |
| Visceral injury | 18 (29) | Out |
| Faecal urgency | 18 (29) | Out |
| Stoma complications | 15 (25) | Out |
| Physical functioning | 12 (19) | Out |
| Readmission | 10 (17) | Out |
| Abandoning the operation | 10 (17) | Out |
| Resection margins | 2 (4) | Out |
| Lymph node yield | 2 (4) | Out |

CIS, core information set; DVT/PE, Deep venous thromboembolism/Pulmonary embolism.

initial long-list of 100 information domains that could be communicated to patients. Established consensus methods were used to prioritise the views of patients and professionals to identify 23 domains of the greatest importance. Finally, consensus meetings with an international group of surgeons and UK patients agreed on the final CIS. It is recommended that the domains in the information set are discussed with all patients before CRC surgery as a baseline to improve patient understanding of expected events and outcomes of surgery and to catalyse questions relevant to the patient for further discussion.

Cancer patients' information needs have been studied extensively.[6 24] While these papers raise important issues, none have examined in detail the information both patients and professionals consider essential to communicate in advance of surgery for CRC. A systematic review identified 239 studies investigating the information needs of patients.[24] It used a qualitative framework analysis to chart patients' documented information needs across

**Box 1    Final core information set**

**Information about experiences in hospital**
► Expected in-hospital experience including length of stay
► Perioperative survival
► Surgical site infection
► Venous thromboembolism
► Anastomotic leak
► Stomata and complications
► Conversion to open operation (where appropriate)

**Information about experiences after discharge**
► Cancer recurrence
► Resection margins
► Long-term survival
► Quality of life including physical and sexual function, faecal incontinence and urgency

studies. 'Treatment-related information' was the most commonly documented need (561 mentions (26%) out of a total 2122 in 239 studies), of which information about 'risks and benefits of treatment' was the most frequent. 'Rehabilitation information' (384 (18%) mentions), specifically 'stoma care' was the next most frequently documented need. A Dutch cross sectional survey asked 101 surgeons (response rate 43%) to rate the importance of 12 selected items of information about risks of anastomotic leak and stomata.[25] Most surgeons reported to 'always' provide information about the risk of anastomotic leak (99%), reoperation (93%) and stoma (93%). It is not clear, however, how this questionnaire was developed, and the scope was limited when compared with the 100 information domains included in this study. Nonetheless, these results triangulate well with the findings of the CIS work presented here.

Although there are no published CISs in CRC surgery, a similar concept has been developed for patients undergoing radiotherapy for rectal cancer.[26] A total of 37 information domains were identified through analysis of audio-recorded clinical consultations and systematic literature reviews. Delphi surveys and consensus meetings with groups of patients and radiation oncologists gained consensus on 13 benefits and harms of treatment that should be discussed in consultations. Some of this information was found to have similar importance with surgeons and patients undergoing CRC surgery including information about local cancer control, survival, sexual and bowel function. Understandably, other information is unique to surgical therapy. Indeed a CIS has been developed for patients undergoing oesophagectomy[12] which shares several characteristics with the colorectal CIS, such as recommending discussions about in-hospital mortality and major complications. Differences in quality of life information reflect the nature of the surgeries, with the colorectal CIS including issues of defecation, whereas the oesophageal CIS recommends discussing eating. In addition, patients and professionals agreed that discussing stomata was important before colorectal but not oesophageal surgery where oesophagostomy is infrequently

required. It is possible that future CIS in other areas of surgical oncology will be developed and generic issues identified across the sets relevant to all patients with cancer.

**Strengths and limitations**

Robust consensus methodology and established guidelines modified from the development of core outcome sets for randomised trials were used to develop this CIS, but there are some weaknesses. The large amount of information identified in phase 1 required the grouping of information into domains. This introduces an element of subjectivity that was minimised through independent dual categorisation; however, information may have been inappropriately grouped or separated. For example, the Dutch CIS for rectal cancer radiotherapy recommended the discussion of five topics around sexual function that this study had combined into one domain.[26] Of note, patients in this study had the opportunity to separate concepts out of domains in the consensus meetings but none chose to do so. Conversely, several domains were amalgamated where participants considered that they were unnecessarily detailed. In addition, patients were involved with the information categorisation process and agreed on the domains in advance of the Delphi process. In phase 2, the scope of the Delphi process was limited to the UK before the CIS development process was opened internationally to professionals in phase 3. This was done to exclude the least important information without the complexity of a multinational Delphi process, however different domains may have been brought forward for discussion at the consensus meetings if this were conducted. It will therefore be important to validate this set in other cultures.

There are no definitive guidelines on sample size and response rates for Delphi studies. The total number of participants in this study is comparatively high and there was a good representation of UK CRC surgical centres, but the response rates from patients was much lower. The effect of this on the validity of the Delphi is unclear because the methodology does not require a representative sample, but to gain a consensus among a wide range of individuals with disparate opinions. In that respect, this study achieved wide diversity based on a priori patient characteristics.

This study has identified an agreed minimum standard of information to be communicated before CRC surgery. Further research is now required to investigate methods to communicate this information effectively in routine practice. This may include agenda setting in clinical encounters,[27] using visual communication aids[28 29] or modifying hospital information leaflets to detail core information. Recent reviews of interventions to improve IC, however, showed that most were inadequately developed, without theory or conceptualisation.[30 31] Included studies were poorly designed, susceptible to bias and outcome measures to assess interventions for IC were inconsistent. It will be necessary to pilot core information carefully, adhering to guidelines for the development of complex interventions,[32] to ensure it is ready for robust evaluation in a cluster

randomised trial. Further research is also needed to understand how CISs relate back to modern professional, ethical and legal standards worldwide.

In conclusion, this study developed an evidence-based CIS to communicate to patients before CRC surgery. It is not intended to replace individualised discussions with patients, but to act as a consistent starting point to catalyse further patient-centred discussions. Core information can be communicated by any professional in any healthcare setting, and form the basis of high-quality written information leaflets. This coordinated and reliable approach to information provision may help patients gain sufficient understanding to undergo surgery for CRC in a post *Montgomery* era.

**Author affiliations**
¹Centre for Surgical Research, Bristol Medical School: Population Health Sciences, University of Bristol, Bristol, UK
²GI Surgery, North Bristol NHS Trust, Bristol, UK
³Centre for Cardiovascular Science, University of Edinburgh, Edinburgh, UK
⁴Division of Surgery, University Hospitals Bristol NHS Foundation Trust, Bristol, UK
⁵Centre for Surgical Research, University of Bristol, Bristol, UK
⁶West of England Academic Health Science Network, Bristol, UK
⁷Cancer Research UK Clinical Trials Unit, Institute of Cancer and Genomic Sciences, University of Birmingham, Birmingham, UK
⁸Population Health Sciences, University of Bristol, Bristol, UK
⁹Consumer Liaison Group, National Cancer Research Institute, London, UK
¹⁰Academic Department of Surgery, University of Birmingham, Birmingham, UK
¹¹Department of Surgery, St Mark's Hospital, London, UK
¹²Department of Academic Surgery, Leeds Institute of Biological and Clinical Sciences, Leeds Teaching Hospitals, Leeds, UK
¹³Centre for Ethics in Medicine, Bristol Medical School: Population Health Sciences, University of Bristol, Bristol, UK
¹⁴Colorectal Site Specific Group, South West Cancer Network, Bristol, UK
¹⁵CSM, University of Oxford, Oxford, UK
¹⁶Department of Surgery, University Hospitals Plymouth NHS Trust, Derriford Hospital, Plymouth, UK
¹⁷Consultant Leeds Institute of Clinical Trials Research, University of Leeds, Leeds, UK

**Acknowledgements** We would like to thank Neil Smart and Katherine Baker for their help running the consensus meetings, and Claudette Blake for her administrative support throughout the whole project.

**Contributors** Wrote the first draft of the manuscript: AGKM RNW RM JR JaB. Contributed to the writing of the manuscript: AGKM RNW RF RM JR BM AP MGT AR DM SD JuB KA SB JaB. Agree with the manuscript's results and conclusions: AGKM RNW RF BM RM JR AP MGT PAS AR AO DM RK DJ RHu RHa SD MGC MC JuB KA SB JaB. All authors have read, and confirm that they meet, ICMJE criteria for authorship.

**Funding** This work was supported by the MRC ConDuCT-II Hub (Collaboration and innovation for Difficult and Complex randomised controlled trials In Invasive procedures - MR/K025643/1) and the NIHR Biomedical Research Centre at the University Hospitals Bristol NHS Foundation Trust and the University of Bristol. RNW was supported by an NIHR doctoral research fellowship. AGKM is funded by a Clinician Scientist Fellowship (NIHR-CS-2017-17-010) from the National Institute for Health Research.

**Disclaimer** The views expressed in this publication are those of the author(s) and not necessarily those of the NHS, the National Institute for Health Research, Health Education England or the Department of Health.

**Competing interests** AGKM is funded by a Clinician Scientist Fellowship (NIHR-CS-2017-17-010) from the UK National Institute for Health Research (NIHR) and supported by the NIHR Biomedical Research Centre at the University Hospitals Bristol NHS Foundation Trust and the University of Bristol.

**Patient consent for publication** Not required.

**Ethics approval** Ethics regulatory approval was granted (UK NRES 10/H0102/82; South West 4 Research Ethics Committee).

**Provenance and peer review** Not commissioned; externally peer reviewed.

**Data availability statement** Data are available in a public, open access repository.

**ORCID iDs**
Angus G K McNair http://orcid.org/0000-0002-2601-9258
Barry Main http://orcid.org/0000-0003-0622-805X
Anne Pullyblank http://orcid.org/0000-0002-2199-1777
Kerry Avery http://orcid.org/0000-0001-5477-2418

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
