## [Reviewer comments · BMJ Open]

ARTICLE DETAILS

TITLE (PROVISIONAL)	The development of a core information for colorectal cancer surgery: a consensus study
AUTHORS	McNair, Angus; Whistance, Robert; Main, Barry; Forsythe, Rachael; Macefield, Rhiannon; Rees, Jonathan; pullyblank, anne; Avery, Kerry; Brookes, Sara; Thomas, Michael; Sylvester, Paul; Russell, Ann; Oliver, Alfred; Morton, Dion; Kennedy, Robin; Jayne, David; Huxtable, Richard; Hackett, Roland; Dutton, Susan; Coleman, Mark; Card, Mia; Brown, Julia; Blazeby, Jane

VERSION 1 – REVIEW

REVIEWER	Patti Shih Australian Centre for Health Engagement Evidence and Values (ACHEEV) School of Health and Society University of Wollongong Australia
REVIEW RETURNED	30-Jan-2019

GENERAL COMMENTS	This is a well-written, clear and coherent paper about the development of Core Information Sets in colorectal cancer treatment. It outlines an extensive methodology designed to design a CIS based on patient-centred shared decision making principles, and report some interesting results of the process itself. I am impressed with the innovation of the methodology, and the detail in which this process is reported and relayed. While I found very few recommendations to make to improve the manuscript, it did occur to me that the authors did not mention how the CIS could be evaluated. Also, given the mentioning of ethical responsibility in patient-communication and the "reasonable-patient", returning to discuss some of these conceptual ideas in the discussion section would better round up the paper. However, these are only minor suggestions which I leave up to the authors to decide if they wish to uptake. I recommend acceptance for publication upon minor editing (eg. some inconsistent abbreviation of "CIS", and the publication status of references No. 19 & 20 may need updating)
--

REVIEWER	Alejandra Recio-Saucedo NIHR Evaluations, Trials and Studies Coordinating Centre, United Kingdom
REVIEW RETURNED	08-Mar-2019

GENERAL COMMENTS

Dear Authors

Thank you for the opportunity to review your manuscript The development of a core information for colorectal cancer surgery: a consensus study, which I read with great interest.

The manuscript reports the development of a Core Information Set (CIS) for patients who are undergoing surgery for colorectal cancer. The research has been reported clearly and the methods are appropriate to reach the aim of the study. Overall, this is an excellent manuscript.

The following notes are recommendations to clarify a couple of aspects of your work and address minor writing style issues found.

- Patient recruitment: What process was followed to invite patients? Where they reached through the centres that took part in the study? Were they identified through patient groups? As indicated, there are no guidelines in terms of minimum number of participants required to validate a Delphi consultation, but I wonder if there would have been opportunities to increase patient participation if different mechanisms to recruit had been used. Maybe this was the case but it is not possible to determine this based on the information available. This information would be of great value to replicate the study with patients undergoing treatments for other diagnoses.

- Sentence on p4 l37-42 "... Patient-led communication, where discussions are guided by the individual, is helpful but patients may lack sufficient baseline knowledge to ask important questions."

There is opportunity to elaborate on the sentence above, given that information and discussions guided by patients are at the core of patient information tools and are key reasons for their existence. When the authors indicate that patients may lack sufficient baseline knowledge to ask important questions, it would be useful to expand the concept of important. Do authors refer here to clinically important questions, linked mainly to the surgical procedure? Do questions also include information that would help patients to make daily-life arrangements for surgery (e.g. length of the surgery; preparations for hospitalisation; clothes to wear after surgery; in this case, concerns about visibility of the stoma) that could be important to the patient but may not feel prepared to ask because they perceive them not to be (clinically) important? On occasion, communication in the clinical encounter may be more influenced by the dynamic between health professionals and patients, than on baseline knowledge. May these questions be important to family or a carer who will provide support to the patient following the surgical procedure? A few more sentences to indicate the support that patient information tools provide to enhance communication between health professionals and patients would strengthen the argument.

Style issues:

P4 l8 Strengths as limitations of this study

P6 l 33 First time use of informed consent abbreviated as IC.

Please define earlier.

P8 l1 ... asked the both ...

	P8 I17 Please rephrase: "Demographic data were collected including area of deprivation" P 9 I31 ... for my less ... P11 I22 This study developed a CIS to use to inform consent ...
--	--

VERSION 1 – AUTHOR RESPONSE

Response to reviewers

Reviewer: 1

Reviewer Name: Patti Shih

Institution and Country: Australian Centre for Health Engagement Evidence and Values (ACHEEV)

School of Health and Society

University of Wollongong

Australia

Please state any competing interests or state 'None declared': None declared

Please leave your comments for the authors below

This is a well-written, clear and coherent paper about the development of Core Information Sets in colorectal cancer treatment. It outlines an extensive methodology designed to design a CIS based on patient-centred shared decision making principles, and report some interesting results of the process itself. I am impressed with the innovation of the methodology, and the detail in which this process is reported and relayed. While I found very few recommendations to make to improve the manuscript, it did occur to me that the authors did not mention how the CIS could be evaluated. Also, given the mentioning of ethical responsibility in patient-communication and the "reasonable-patient", returning to discuss some of these conceptual ideas in the discussion section would better round up the paper. However, these are only minor suggestions which I leave up to the authors to decide if they wish to uptake. I recommend acceptance for publication upon minor editing (eg. some inconsistent abbreviation of "CIS", and the publication status of references No. 19 & 20 may need updating)

Thank you. We outlined methods to evaluate CIS on paragraph 2, page 14 including examples of how CIS could be introduced into clinical practice and the use of complex intervention development methodology. Revisions have been made to the abbreviations and references 19 and 20.

Reviewer: 2

Reviewer Name: Alejandra Recio-Saucedo

Institution and Country: NIHR Evaluations, Trials and Studies Coordinating Centre, United Kingdom

Please state any competing interests or state 'None declared': None declared

Please leave your comments for the authors below

Dear Authors

Thank you for the opportunity to review your manuscript The development of a core information for colorectal cancer surgery: a consensus study, which I read with great interest.

The manuscript reports the development of a Core Information Set (CIS) for patients who are undergoing surgery for colorectal cancer. The research has been reported clearly and the methods are appropriate to reach the aim of the study. Overall, this is an excellent manuscript.

Thank you

The following notes are recommendations to clarify a couple of aspects of your work and address minor writing style issues found.

- Patient recruitment: What process was followed to invite patients? Where they reached through the centres that took part in the study? Were they identified through patient groups? As indicated, there are no guidelines in terms of minimum number of participants required to validate a Delphi consultation, but I wonder if there would have been opportunities to increase patient participation if different mechanisms to recruit had been used. Maybe this was the case but it is not possible to determine this based on the information available. This information would be of great value to replicate the study with patients undergoing treatments for other diagnoses.

The recruitment process involved identifying eligible patients through clinical record reviews at participating clinical centres. Invitations were sent through the post including a cover letter, participant information leaflet and consent form. A pre-paid return addressed envelope was provided for the completed consent form for those who wanted to participate. This has been clarified in the manuscript on page 7.

It is agreed that other opportunities may have increased patient participation. For example, national patient support charities may keep records of patients willing to take part in research. Recruitment through these group, within appropriate governance protocols, may be superior.

- Sentence on p4 l37-42 "... Patient-led communication, where discussions are guided by the individual, is helpful but patients may lack sufficient baseline knowledge to ask important questions."

There is opportunity to elaborate on the sentence above, given that information and discussions guided by patients are at the core of patient information tools and are key reasons for their existence. When the authors indicate that patients may lack sufficient baseline knowledge to ask important questions, it would be useful to expand the concept of important. Do authors refer here to clinically important questions, linked mainly to the surgical procedure? Do questions also include information that would help patients to make daily-life arrangements for surgery (e.g. length of the surgery; preparations for hospitalisation; clothes to wear after surgery; in this case, concerns about visibility of the stoma) that could be important to the patient but may not feel prepared to ask because they perceive them not to be (clinically) important? On occasion, communication in the clinical encounter may be more influenced by the dynamic between health professionals and patients, than on baseline knowledge. May these questions be important to family or a carer who will provide support to the patient following the surgical procedure? A few more sentences to indicate the support that patient information tools provide to enhance communication between health professionals and patients would strengthen the argument.

The theory supporting the development of the CIS is derived from prominent ethicists Faden and Beauchamp. They argue a definition of informed consent, known as "autonomous authorisation", that occurs when an individual with "substantial understanding" of the nature of an action and the foreseeable consequences and possible outcomes intentionally authorises the action. Faden and Beauchamp propose that "substantial understanding" involves the "apprehension of all the material or important descriptions" of an action. It is therefore differentiated from "full understanding" in that a person could be ignorant of relevant but "unimportant" information. "Important" information is subjective in that it is enough that an individual views information as important to make it important, and what is important to one person may not be to another.

This moral theory creates problems when considering standards of disclosure. Any developed standard of disclosure appears to be inappropriate, as distinguishing important from unimportant information on an individual basis would be difficult. Although Faden and Beauchamp accept the subjective view of substantial understanding, they argue that this does not completely negate the need for the traditional patient disclosure.

"In the absence of a disclosure initiated by the professional, how could patients or subjects, who often know virtually nothing about the object of choice, begin to formulate their concerns, let alone ask meaningful questions?"

The purpose of the CIS is to address this baseline information, and methods focussed on identifying what stakeholders considered "important". We did not specify what information was relevant a priori

other than framing the research question in the context of someone having colorectal cancer surgery. Information that could be considered irrelevant by some, such as location of the hospital, was identified and considered important through the domain “patient pathway”.

It is acknowledged that patient encounters are more complex than presented here and includes dynamics with family, friends and others. Developing a CIS for informed consent, as opposed to another concept such as shared decision making, aimed to simplify clinical encounters precisely because it represents the authorisation of an action by an individual autonomous person. That person may consider the views of others when formulating what they consider “important”, as did our research participants, but it is ethico-legally the prerogative of the individual.

This has been clarified in the manuscript on page 4 and 7.

Style issues:

P4 I8 Strengths as limitations of this study Done

P6 I 33 First time use of informed consent abbreviated as IC. Please define earlier. Done

P8 I1 ... asked the both ... Done

P8 I17 Please rephrase: “Demographic data were collected including area of deprivation” Done

P 9 I31 ... for my less ... Done

P11 I22 This study developed a CIS to use to inform consent ... Done

VERSION 2 – REVIEW

REVIEWER	Alejandra Recio Wessex Institute, University of Southampton, UK
REVIEW RETURNED	11-Apr-2019

GENERAL COMMENTS	Dear Mr McNair and Authors Thank you for your responses and revision of the manuscript: “The development of a core information for colorectal cancer surgery: a consensus study.” The information added to various sections in the paper has clarified the questions raised and appropriately addressed the changes suggested. It was especially interesting to read your reflection on Faden and Beauchamp which would be a great foundation for a discussion on the complexities of effective, informed and educated consent, autonomous decision-making, and genuine autonomous choices. The outputs of your study would be of great interest to researchers developing information tools for patients; those exploring the role of effective consent in medical treatment decisions; as well as researchers interested in the Delphi method. Very best wishes
---